# Evaluation of Non-Contact Device to Measure Body Temperature in Sheep

**DOI:** 10.3390/ani14010098

**Published:** 2023-12-27

**Authors:** Carla Ibáñez, María Moreno-Manrique, Aránzazu Villagrá, Joel Bueso-Ródenas, Carlos Mínguez

**Affiliations:** 1Department of Animal Production and Public Health, Faculty of Veterinary Medicine and Experimental Sciences, Catholic University of Valencia “San Vicente Mártir”, 46001 Valencia, Spain; carla.ibanez@ucv.es (C.I.); cminguez.balaguer@ucv.es (C.M.); 2Doctoral School, Catholic University of Valencia “San Vicente Mártir”, 46001 Valencia, Spain; maria.morenomanrique@gmail.com; 3Centro de Investigación en Tecnología Animal (CITA), Valencian Institute for Agricultura Research (IVIA), 12400 Segorbe, Spain; villagra_ara@gva.es

**Keywords:** rectal thermometry, infrared thermography, thermal imaging, infrared thermometers, manchega ewe

## Abstract

**Simple Summary:**

Rectal thermometry is still standard practice in the measurement of body temperature in livestock because of its low cost. However, there are situations where this method is contraindicated or impractical, and a less invasive method is required. Non-contact devices have been used in livestock production; however, there are few studies about the variation and correlations in body temperature between rectal temperature and non-contact devices. The present work aims to compare rectal temperature with two types of non-contact devices (non-contact infrared thermometers and thermal imaging/infrared thermography) for the assessment of body temperature in healthy sheep. Except for the temperature taken by non-contact infrared thermometers at the muzzle, the correlation between rectal temperature vs. non-contact infrared thermometers or thermal imaging/infrared thermography showed a low significance or was difficult to use for practical flock management purposes. In addition, the variability between devices was high, which implies that measurements should be interpreted with caution in warm climates and open pens, such as most sheep farms in the Spanish Mediterranean area.

**Abstract:**

Non-contact devices have been used in the measurement of body temperature in livestock production as a tool for testing disease in different species. However, there are few studies about the variation and correlations in body temperature between rectal temperature (RT) and non-contact devices such as non-contact infrared thermometers (NCIT) and thermal imaging/infrared thermography (IRT). The objective of this work was to evaluate the accuracy of non-contact devices to measure the body temperature in sheep, considering six body regions and the possibility of implementing these systems in herd management. The experiment was carried out at the experimental farm of the Catholic University of Valencia, located in the municipality of Massanassa in July of 2021, with 72 dry manchega ewes, and we compared the rectal temperature with two types of non-contact infrared devices for the assessment of body temperature in healthy sheep. Except for the temperature taken by NCIT at the muzzle, the correlation between RT vs. NCIT or IRT showed a low significance or was difficult to use for practical flock management purposes. In addition, the variability between devices was high, which implies that measurements should be interpreted with caution in warm climates and open pens, such as most sheep farms in the Spanish Mediterranean area. The use of infrared cameras devices to assess body temperature may have a promising future, but in order to be widely applied as a routine management method on farms, the system needs to become cheaper, simpler in terms of measurements and quicker in terms of analyzing results.

## 1. Introduction

Measurement of body temperature is one of the main tests routinely performed in the physical examination of an animal in veterinary medicine [1]. In sheep, it can be used as an indicator for the early detection of diseases [2]. An increase in animal body temperature could indicate a possible viral infection (bovine viral diarrhea or foot and mouth disease), an inflammatory disease (laminitis or mastitis) or a measure of stress [3]. Additionally, this parameter has economic importance, as sheep with hyperthermia or hypothermia are animals that are sluggish, spend more time lying down and decrease feed intake, resulting in a decrease in milk production [4].

Rectal thermometry is still standard practice in the measurement of body temperature in livestock because of its low cost. However, there are situations where this method is contraindicated or impractical and a less invasive method is required. Additionally, the operation process can make animals produce stress reactions, resulting in a larger measurement error [5]. To all this it should be added that one of the main problems with rectal temperature (RT) testing is the possible risk of transmitting diseases, especially when a considerable number of animals have to be tested in a short period of time [6].

In recent years, new methods have been developed to try to measure body temperature in a non-invasive way. Infrared systems are a technology that allows the measurement of heat transfer and changes in blood flow without contact with the animal and without the need for restriction of movement [7]. The aim of these technologies is the early detection of heat stress, disease or illness in real time that minimizes errors and reduces labor cost [8]. Additionally, infrared systems offer a safe means to measure body temperature without physical and zoonotic risk for either the animal and the farmer, since it can be done remotely [9].

Thermal imaging/infrared thermography (IRT) has been used in livestock production as a tool for testing disease in horses, beef and dairy cattle or rams [10]. This non-invasive technology allows farmers to monitor and evaluate the surface temperature of animals efficiently and swiftly. The ability to assess numerous animals simultaneously provides a comprehensive overview of the herd’s thermal status, allowing for early detection of potential health issues and a timely intervention. This technology proves particularly beneficial in large-scale farming operations, where a prompt and accurate assessment of the thermal comfort of a substantial number of animals contributes to improved animal welfare, enhanced productivity, and efficient resource allocation [11,12,13]. It should be noted that the region of the body where IRT is performed will differ between species, as different degrees of correlation with RT and ambient temperature have been observed [14]. The most commonly used regions to determine disease or stress by IRT are the udder, body, side, hooves, ears, muzzle, eye or neck, and the vulva for reproduction traits [15]. However, there are few studies about the variation and correlations in body temperature between RT and non-contact devices (non-contact infrared thermometers (NCIT) and IRT).

The objective of this work was to evaluate the accuracy of non-contact devices in measuring the body temperature in sheep while considering different areas, such as the lachrymal caruncle (LC), vulva (VL), muzzle (MUZ), left ear (EA), neck (NE) and back leg (BL), and the possibility of implementing these systems in herd management.

## 2. Materials and Methods

The protocol used for sheep handling was approved by the Committee of Ethics in Animal Experimentation of the Catholic University of Valencia with the code CEEAUCV2008.

### 2.1. Animals

The experiment was carried out at the experimental farm of the Catholic University of Valencia, located in the municipality of Massanassa, in July of 2021. Dry manchega ewes (*n* = 72) were intensively reared in a well-ventilated and straw-bedded barn without access to pasture. Animals were sheared in May of the same year, as per usual in sheep herds of this geographical area. The fodder used consisted of cereal straw, maize cane silage and artichoke by-product. Additionally, animals received a commercial feed supplement of 1000 g per head of cereal mixture (Cebicor^®^, de Heus, corn 20%, barley 19%, oat groats 18%, dried beet pulp 17%, soybean meal 6.2%, peas 8%, sunflower seeds 3.3%, sunflower meal 3%, dried orange pulp 4.8%, salt 0.4% and soybean oil 0.3%) and were provided with feed in the first hour of the morning. All animals used in the study were examined by a veterinarian and were in apparent good health. Those animals presenting fever, respiratory problems and/or diarrhea were removed from the experiment.

### 2.2. Temperature Measurements

For each animal, the LC, VL, MUZ, EA, NE, BL (both for a NCIT and IRT) and RT were measured simultaneously. One single operator was responsible for taking the data for each of the measurement systems. The time required to take all measurements was less than 3 min. Temperature measurements were taken on the same day for all animals, starting at 08:00 and ending at 14:00. Ambient temperature (°C) and relative humidity (%) were evaluated every 15 min at the place where the measurements were taken by using a Testo 174 h datalogger (Barcelona, Spain). The average data were 28 ± 3 °C and 65 ± 5% RH, respectively.

At the start of the experiment, feed was served and the sheep were trapped by closing the headlock in groups of 5 animals, thus minimizing the time that the animals were head-locked, which was always less than 15 min. Before taking the temperature, the area of incidence was previously cleaned. To measure RT, a digital thermometer SC 12 (Hauptner and Herberholz, Solingen, Germany) was inserted 3 cm into the rectum. Thermometer IR300UV (Extech Instruments Ltd., Waltham, MA, USA) was used for NCIT. Emissivity was set to 0.95, as recommended by the maker for animal body surfaces. The accuracy of this device is ±1.5 °C or 1.5% of the reading when the environmental temperature is between 0 and 33 °C and the thermal sensitivity is 0.1 °C. This thermometer has LED guide lights to indicate the area and distance at which the temperature is measured. The temperature was noted when it stabilized (10–15 s approximatively). A Testo 868 infrared thermography camera (Barcelona, Spain) was used for IRT. This camera has a built-in digital camera, image quality with IR resolution of 160 × 120 pixels, thermal sensitivity from 0.08 °C, and the emissivity was set at 0.95. The methodology of use is based on placing the camera lens part in the direction of the region to be studied and, once the image is visualized, taking a photo. This method was always carried out by taking the same area in each named body region, with an approximate distance of about 25 cm from the animal’s body and practically a right angle to the region to be measured (Figure 1a,b). The infrared thermography camera images were processed with the software testo IRSoft v5.0 (Barcelona, Spain) to adjust the ambient temperature and relative humidity for a better measurement of the temperatures of the body surfaces of the sheep.

### 2.3. Statistical Analysis

The software R Project program version 4.1.2 [16] was used to evaluate the relationship between the thermometer type and incidence area of NCIT and IRT. Correlations between RT, NCIT and IRT were determined using the “corrgram” package [17]. Correlations with coefficients > 0.75 were classified as high, those between 0.36 and 0.70 as moderate, and those <0.35 as low [18]. GLM procedures were used to estimate the effect of the incidence location and devices. A level of significance was established at α = 0.05.

## 3. Results

Table 1 shows the summary statistics for the measured traits. When comparing the three devices, RT had the highest value. For the temperatures taken by the NCIT and IRT devices, the same trend is observed for each location. The BL temperature showed lower values together with a higher variability, and the VL showed higher values with a lower S.D.

Figure 2 shows the values of the correlations between the measured parameters as well as the level of significance (*** = *p*-values < 0.0005; ** = *p*-values < 0.005; * = *p*-values < 0.05; NS = not significant). High positive correlations are observed between RT and NCIT_ MUZ (*r* = 0.78); NCIT_EA and IRT_BL (*r* = 0.78); NCIT_VL and NCIT_BL (*r* = 0.86); IRT_EA and IRT_BL (*r* = 0.79), with the correlation between NCIT_EA and IRT_EA standing out (*r* = 0.91). A high negative correlation was found between NCIT_MUZ and IRT_VL (*r* = −0.73).

Table 2 reports the least squared means ± SD of temperatures depending on the location and the device used. RT was significantly higher compared with non-contact devices. IRT showed higher significant values than NCIT, with an average difference (considering all locations) of 2.34 °C. Additionally, the locations of the highest to lowest temperature (same trend in both devices) were VL, LC, NE, MUZ and BL, respectively. All locations had significant differences between them except for MUZ vs. NE and LC vs. VL, which showed equal results (in both devices). Comparing the different thermometers used, the rectal was found to have the least bias (considering all regions investigated).

## 4. Discussion

The main objective of the study was to correlate RT with some non-invasive measurements, which could be widely used by farmers on a practical level. It should be noted that IRT devices are expensive and require specific software, a thermographic image formation and precautions in obtaining thermograms to avoid errors in the production of images. This last subject could lead to biases in the interpretation of results and lead to diagnostic failures if the technique is not properly performed [19]. This makes it a method that is increasingly studied and developed in research or precision livestock farming [20]. For this reason, at the level of daily herd management by farmers, a NCIT device could be more interesting, as it is simpler and cheaper [21]. Furthermore, the RT measurement requires a minimum of 15 s per measurement as opposed to the 5 s required by the NCIT [22]. This time is increased due to the process of cleaning the rectal thermometer after measuring each animal to avoid risks of cross-contamination [5]. 

Referring to the summary data, the highest temperature values showed the lowest variance. Many studies estimate this relationship because they are less influenced by the dirt of the animals [23,24].

Most studies made on animals agree with our results stating that RT is higher than all other temperatures taken with IRT or NCIT, which is logical given that it is measured incorporeally [1,3,6,10,23,24,25,26]. IRT and NCIT measured in EA was strongly correlated (*r* = 0.91). This result makes sense, as the inner surface of the ear is a uniform area in the sense that it is not wet and does not contain hair, dung, crusts or wounds. These factors can impede the propagation of infrared waves and can also contribute to evaporative heat loss [27,28,29]. An interesting correlation on a practical level is the one between RT and NCIT MUZ (*r* = 0.78), since it can be easily adaptable to operating conditions on a farm in production. This high correlation is probably due to the absence of a fat layer and thinner skin without hair [22]. The remaining correlations calculated in the present study show that not all the techniques and devices used are suitable for generating reliable and accurate results relating RT to NCIT or IRT. Our results agree with studies that show moderate or low correlations between the different devices used and measurement locations [28,30,31]. However, IRT and NCIT have been used in several studies as a standard technique to measure body temperature in other ruminants, despite moderate correlations between both values being found in dairy cows [32] or in beef cattle [33]. As seen in the present study and others developed in piglets [22] or in humans [34,35], hairless skin or eyes are good area choices to measure body temperature. However, despite the high correlations found in some studies also developed in sheep [36], it is important to find practical ways to measure the body surface temperature, given that despite the highest correlations being found in the mentioned study in the inner sides of the arms and thighs of the animals, these measurements would imply animal handling, involving less practicality and more time. A comparison between studies is complicated by differences in the experimental group size, species (including breeds) [37,38], distance between the device and the animal [22,39] or operator effects [40]. However, without a doubt, the effects that can contribute the most differences are the different environmental conditions [19]. Factors such as humidity, the air speed, direct sunlight or ambient temperature can be decisive in obtaining values in NCIT and IRT devices [5,10,41,42].

The RT range obtained in the study is within the normal temperature for sheep [43]. This study was developed in a Mediterranean wet region during a warm season. In response to elevated ambient temperatures, sheep employ a multifaceted approach to thermoregulation to maintain their core body temperature within a narrow, optimal range. When faced with heat stress, sheep primarily rely on behavioral adaptations such as seeking shade and avoiding direct exposure to the sun [44]. Additionally, they engage in increased respiration as a means of dissipating excess body heat [45,46]. The intricate interaction between physiological responses, such as vasodilation to enhance heat loss through the skin, and an increased respiration rate underscores the adaptability of sheep to challenging thermal conditions. Understanding these nuanced mechanisms is crucial for effective management practices to mitigate heat stress impact on sheep welfare and productivity in warmer climates. In the present study, the environmental temperature was 28 ± 3 °C. In these conditions, well-adapted animals, such the manchega breed sheep employed in this study, have the capability to dissipate excess body heat by increasing their respiration rate and vasodilation. Thus, IRT and NCTI of the vulva and ear temperature showed high values, and the muzzle radiated temperature was lower, while the rectal temperature was maintained at physiological values [43,47].

Considering Table 1 and comparing all measurements, RT showed the smallest deviation, which is why it remains the reference standard for measuring the core body temperature [23,25]. Between NCIT and IRT, measurements showed the same deviation, so this indicates that the surface temperature measurement can be described with equal accuracy in both methods [42]. It should be noted that the results of our study show discordances between infrared devices as well as in different zones within the same device. This could be considered as results that are neither reproducible nor constant. 

## 5. Conclusions

In the present study, we compared the rectal temperature with two types of non-contact infrared devices for the assessment of body temperature in healthy sheep in six body regions. Except for the temperature taken by NCIT at the muzzle, the correlation between RT vs. NCIT or IRT showed a low significance or was difficult to use for practical flock management purposes. In addition, the variability between devices was high, which implies that measurements should be interpreted with caution in warm climates and open pens, such as most sheep farms in the Spanish Mediterranean area.

The use of infrared camera devices to assess body temperature may have a promising future, but in order to be widely applied as a routine management method on farms, the system needs to become cheaper, simpler in terms of measurements and quicker in terms of analyzing results.

## Figures and Tables

**Figure 1 animals-14-00098-f001:**
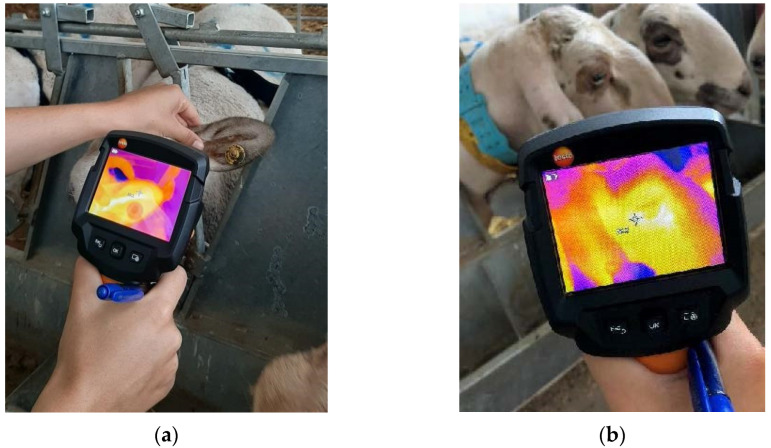
Testo 868 infrared thermography camera (Barcelona, Spain). (**a**) EA measurement; (**b**) LC measurement.

**Figure 2 animals-14-00098-f002:**
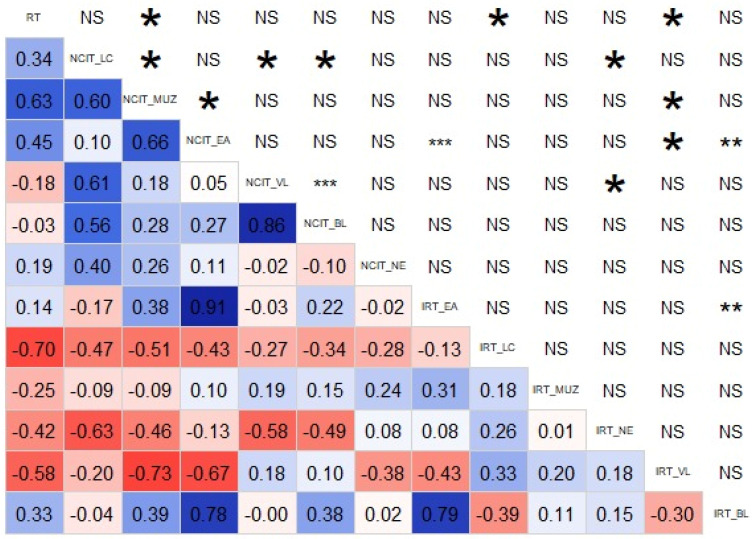
Correlation matrix between measured parameters. Upper triangle corresponds to the significance level (*** = *p*-values < 0.0005; ** = *p*-values < 0.005; * = *p*-values < 0.05; NS—not significant). Lower triangle corresponds to the values of the correlations. Diagonal shows the parameters (RT—Rectal thermometer, NCIT—Non-contact infrared thermometers, IRT—Thermal imaging/infrared thermography, LC—lachrymal caruncle, VL—vulva, MUZ—muzzle, EA—ear, NE—neck, BL—back leg). Blue color corresponds to positive correlations and red to negative ones. The intensity of the color depends on the value of the correlation, being fainter close to zero.

**Table 1 animals-14-00098-t001:** Summary statistics for temperature (°C) measured by the different devices in each region of incidence.

**Temperature Device**	**Region**	**Mean**	**S.D.**	**Minimum**	**Maximum**
RT	Rectum	38.88	0.49	37.90	40.10
NCIT	LC	34.86	1.10	32.50	37.40
NCIT	MUZ	33.45	1.12	30.40	36.30
NCIT	EA	31.40	2.28	26.80	35.30
NCIT	VL	35.12	0.88	32.50	37.50
NCIT	NE	33.83	0.94	31.50	33.90
NCIT	BL	30.28	1.70	26.60	33.90
IRT	LC	36.97	1.01	33.80	39.00
IRT	MUZ	35.49	1.07	32.80	37.70
IRT	EA	33.76	2.30	29.50	37.90
IRT	VL	38.27	0.84	34.90	39.70
IRT	NE	36.00	1.17	33.10	38.30
IRT	BL	33.64	1.46	30.20	36.00

S.D.—Standard deviation, RT—Rectal thermometer, NCIT—Non-contact infrared thermometers, IRT—Thermal imaging/infrared thermography, LC—lachrymal caruncle, VL—vulva, MUZ—muzzle, EA—left ear, NE—neck, BL—back leg. Total observations: 936.

**Table 2 animals-14-00098-t002:** Least square means (±standard error) of temperature (°C) obtained with the different devices and the location of measurement.

Location	Digital	NCIT	IRT
Rectum	38.88 (0.26)	-	-
Back leg	-	30.8 (0.14) ^a#^	33.1 (0.14) ^a^*
Ear	-	31.5 (0.14) ^b#^	33.8 (0.14) ^b^*
Lachrymal caruncle	-	34.7 (0.14) ^c#^	37.1 (0.14) ^c^*
Muzzle	-	33.3 (0.14) ^d#^	35.6 (0.14) ^d^*
Neck	-	33.7(0.14) ^d#^	36.1(0.14) ^d^*
Vulva	-	35.0 (0.13) ^c#^	37.3 (0.13) ^c^*

^abcd^ Means in the same column with the same superscript do not differ significantly. ^#^* Means in the same row with the same superscript do not differ significantly. Significant difference at α = 0.05.

## Data Availability

Data are contained within the article.

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
