# Peer review of "Evaluation of Non-Contact Device to Measure Body Temperature in Sheep"

_animals, 2023, doi:10.3390/ani14010098_

Round 1

Reviewer 1 Report

Comments and Suggestions for Authors

Animal 2740909. C. Ibanez et al

General comments: This is an interesting study and provides information of use to the sheep industry. The authors would be encouraged to revise the manuscript in view of the following comments and resubmit.

Specific Comments:

The authors imply that the use of core temperatures (rectal) remains the biometric of choice for determining an animal’s temperature. The use of rectal temperature probes is simply not a practical solution for ever increasing farms of scale in animal agriculture. The authors would be encouraged to revise their manuscript in view of more recent reviews on this matter example Advances in precision livestock farming Ed By D. Berckmans. Burleigh Dodds. 2022.

Methodology concerns

1. -there is no mention of the state of shearing! This will cause variation in heat management among the animals.

2. -the temperatures for the sheep are high (38 C plus) – please describe in more detail the state of herd health

3. -the sheep appear to be hyperthermic and are thermoregulating to remove heat ( high vulva and ear temp) hence, they are not thermoneutral and in steady state. Also suggested by high vulva and low muzzle radiated temperature.  In that situation there will not be good correlation between core and radiated because the sheep are using radiated temperature loss to thermoregulate.

4. -What was the respiration rate? This value would help to clarify hyperventilation to > respiratory heat loss.

5. -the sheep are restrained for the measurements – restraint is a significant stressor for the animals which will also cause variation in thermal temperature.

6. -the measurements appear to have taken several hours to collect during which time the atmospheric temperature (using the SD values to calculate) would have varied by upwards of 10 C. That will significantly affect the radiated temperature yet no correction for this factor appears to have been made.   It is essential that the radiated temperatures in particular be corrected for variation in environmental temperature.

7. -the Testo and Extech instruments are comparatively low  precision. Also, no data is provided for calibration factors and again, correction for environmental variation.

8. -the radiated instrument is reported to use auto emissivity! That is not a reliable feature in a radiation monitoring device. Emissivity needs to be set properly.

9. Minor – L11 -  change their to its?

10. -As a result of the above

The authors are misinterpreting the fundamental components of thermoregulation and the role different mechanisms have in controlling an animals homeothermic condition.

A very basic analogy would be to think about a car motor. It produces heat constantly and the amount of heat will vary depending on whether the car is changing speed, driving through varied terrain, etc. Accordingly, the radiator of the car will constantly be opening the thermostat to increase heat loss or closing the thermostat to increase heat retention. All this time the core engine temperature will remain constant. Hence, of course the correlation between the core engine temperature and the radiator temperature will vary and at times be well correlated yet at other times be quite different.

The same is true for a homeothermic animal. The animal, in this case a sheep, will try to keep a constant core temperature. However, events in its life such as the time of day, temperature of the environment, time since feeding and the incremental heat of eating, variation in wool cover, variation in stress susceptibility to the stress of restraint and so forth will cause the animal to use its “radiator” namely radiated heat loss, to control the core temperature.  Between 40-60% of the energy loss in a homeothermic mammal is through radiated infrared thermal loss. Hence, for an animal, there will be times the core and radiated temperature are well correlated and there will be times the correlation is not high. This does not imply one methodology is better or worse or more or less accurate than another.

The authors would be encouraged to reflect  and incorporate these factors in a revised manuscript.

11. The reference list could be expanded somewhat to include manuscripts reflecting the above points.

12. It is a fair amount of work to monitor some seventy animals for these measurements. The authors are to be commended for their effort.

Comments on the Quality of English Language

The quality of English is very good

Reviewer 2 Report

Comments and Suggestions for Authors

General comment:

The present manuscript evaluates the effectiveness of non-contact infrared devices in measuring body temperature. While recent research has explored this technology, its application in sheep remains relatively unexplored. The study's findings are valuable for farmers and industry professionals focused on animal welfare. The study design was straightforward, and the manuscript was easy to follow. I have a couple of minor suggestions listed below.

The introduction was well-constructed and succinctly justified the study's necessity.

Line 106: into de rectum?

Line 109: seg approximatively?

Could you please specify the total number of observations? Was it 72 for each region?

Line 125–126: consider relocating the sentence 'Discussion' section.

Line 135–138: Consider relocating the classification of correlations within the 'Statistical Analysis' subsection under the 'Materials and Methods' section,

Line 167: Consider revising the sentence for improved clarity.

The results and discussion sections were well-presented and easy to follow. The manuscript appropriately acknowledged the study's limitations, which adds to its credibility.

The conclusion aligns well with the study objectives and accurately reflects the study's findings.

Comments on the Quality of English Language

Please consider rephrasing a couple of sentences as mentioned above.

Reviewer 3 Report

Comments and Suggestions for Authors

The paper is clearly written and easy to understand. However, the results are difficult to read.

With regard to Table 1, I suggest separating the temperature device from the area of the animal where the measurement was taken. In other words, turn the first column into two - the temperature device and the region of the animal where the measurement was taken.

The methodology does not make it clear whether or not the animals were kept in headlocks throughout the entire data collection period.
